# Population Assessments of Federally Threatened Everglades Bully in Big Cypress National Preserve, Florida, USA, Using Habitat Suitability Modeling and Micromorphology

**DOI:** 10.3390/plants12071430

**Published:** 2023-03-23

**Authors:** James J. Lange, Courtney L. Angelo, Erick Revuelta, Jennifer Possley

**Affiliations:** 1Smart-Sciences, Inc., 330 SW 27th Ave STE 504, Miami, FL 33135, USA; jlange@smart-sciences.com; 2Fairchild Tropical Botanic Garden, 10901 Old Cutler Rd., Miami, FL 33156, USA; 3Big Cypress National Preserve, 33100 Tamiami Trail E, Ochopee, FL 34141, USA; 4St. John’s River Water Management District, 4049 Reid Street, Palatka, FL 32177, USA

**Keywords:** habitat suitability modeling, Everglades bully (*Sideroxylon reclinatum* subsp. *austrofloridense*), Big Cypress National Preserve scanning electron microscope, plant conservation, cryptic speciation

## Abstract

In Big Cypress National Preserve, the federally threatened Everglades bully (*Sideroxylon reclinatum* subsp. *austrofloridense*) is sympatric with its conspecific, more widespread relative, the Florida bully (*Sideroxylon reclinatum* subsp. *reclinatum*). In this area of overlap, the only reliable characters to distinguish the two are cryptic, micromorphological traits of the abaxial laminar surface. In order to better understand the distribution of the federally threatened taxon, we used a combination of habitat suitability modeling (HSM), field surveys, and microscopy. Using models to inform initial surveys, we collected leaf material of 96 individuals in the field, 86 of which we were able to identify to subspecies. Of these, 73 (85%) were identified as the threatened taxon, expanding both the known range and population size within Big Cypress. We used these 73 new occurrences to rerun HSMs to create a more accurate picture of where the taxon is likely to occur. A total of 15,015 hectares were predicted to be suitable habitat within Big Cypress, with 34,069 hectares across the entire study area. These model results could be used to inform the critical habitat designation for this taxon. For at-risk, cryptic taxa, such as the Everglades bully, multiple approaches are needed to inform management and conservation priorities, including the consideration of a hybridization zone.

## 1. Introduction

A foundation of organismal conservation is an understanding of the rarity of a given taxon. Rarity generally refers to a taxon’s distribution and abundance [1], but can include other factors such as level of habitat specificity [2]. The process of quantifying rarity is inextricably linked to the discipline of taxonomy, which seeks to first define the entity of concern, be it species or intraspecific taxon, as only then can rarity be assessed [3]. Unique taxa are not always discernible using morphological characters alone, despite being evolutionarily distinct based on other criteria, a phenomenon known as cryptic speciation [4]. Advancements in genetic techniques along with a focus on behavioral ecology and various micromorphologies are increasingly identifying new taxa which do not have a clear morphological distinction, i.e., cryptic species [5,6,7]. These advancements in taxonomic research are providing new tools for understanding rarity, thus enabling a more complete assessment of threats and conservation priorities that can improve our ability to save the most at-risk taxa. 

Morphology of laminar surfaces can be particularly informative, as these organ characters can be highly polymorphic and can generate distinguishable features between taxa. Venation and trichome characters are recognized widely as taxonomic tools, but less emphasis has been placed on micromorphological characters such as stomata and epidermal cell walls, likely due to the difficulty and cost of examination. However, this approach has been found to be instrumental in determination of taxa within several widespread genera such as *Solanum* L., *Persicaria* Mill., and *Crotalaria* L. [8,9,10], to name a few.

The purpose of this study was to determine the extent of an at-risk, cryptic taxon within Big Cypress National Preserve (BICY) in South Florida, USA. The Everglades bully (*Sideroxylon reclinatum* Michx. subsp. *austrofloridense* (Whetstone) Kartesz and Gandhi) was first described in 1985 [11] and was listed as federally threatened in 2017 [12]. *S. reclinatum* sensu lato (s.l.) is a woody shrub in the Sapotaceae found throughout Florida and portions of the Southeastern Coastal Plain. The subspecies was recognized on the basis of abaxial laminar surfaces, pedicels, and calyx being consistently rufous-tomentose, while these on the wider-ranging subsp. *reclinatum* were glabrous or with scattered trichomes along the abaxial midvein [11] (Figure 1). At the time of listing, subsp. *austrofloridense* was known only from Miami-Dade County, FL, chiefly within Everglades National Park (EVER) but also in a limited number of Miami-Dade County and South Florida Water Management District preserves, in habitats including marl prairie, pine rockland, and prairie/pine rockland ecotone. Subsp. *austrofloridense* was not documented in BICY until 2002, when it was discovered during a plant inventory by The Institute for Regional Conservation (IRC) [13]. IRC did not provide detailed population numbers for the taxon in BICY at the time since individual taxa were not the focus of the study. However, subsequent surveys by IRC in 2013 within the Lostmans Pines region of BICY provided a baseline for population estimates [14]. The authors discussed difficulties in identifying *S. reclinatum* s.l. to subspecies during their surveys. Many individuals displayed laminar pubescence characters intermediate between the widespread subsp. *reclinatum* and the South Florida endemic subsp. *austrofloridense*. IRC conservatively determined individuals to be subsp. *austrofloridense* only if mature leaves displayed conspicuous pubescence throughout the abaxial surface. By this standard, they documented 17 individuals of subsp. *austrofloridense* in the Lostmans Pines region. 

Shortly thereafter, Corogin and Judd [15] published a detailed analysis of the two subspecies using micromorphological characters of abaxial leaf surfaces, which they considered to be more reliable than pubescence. This approach had previously been applied by Anderson [16] in distinguishing the rare *S. thornei* (Cronquist) T.D.Penn. from superficially similar congeners. Corogin and Judd’s analysis revealed that both subspecies are cryptically sympatric in BICY. Of five (5) specimens examined from the southern portion of BICY, three (3) were determined through micromorphology to be subsp. *austrofloridense*, confirming the need for a more detailed and precise survey of the species complex in BICY. 

Given the relatively recent discovery of subsp. *austrofloridense* within BICY, along with the vast unsurveyed marl prairie/pineland ecotones found within the Preserve, a more thorough evaluation of the taxon’s distribution needed to be explored. For this purpose, we developed habitat suitability models (HSMs) for the taxon, which were used to inform surveys of potential suitable habitat in early 2022. HSMs generated locations for novel areas to survey under this study, both to expand our knowledge of subsp. *austrofloridense* in BICY, but also to survey for optimal translocation sites for taxon restoration, if deemed necessary. We then followed protocols developed by Corogin and Judd [15] to identify a subset of individuals to subspecies and used newly expanded occurrence data to update the HSMs. This expanded spatial dataset will help to inform resource management strategies based on the current condition and rarity of subsp. *austrofloridense*, along with the amount of potential suitable habitat for the taxon. 

## 2. Results

### 2.1. Model Results

For reference, area under the curve (AUC) scores above 0.9 indicate high accuracy, scores between 0.7 and 0.9 indicate useful applications, and values of 0.5 to 0.7 indicate low accuracy [17].

#### 2.1.1. Pre-Survey Models

The pre-survey model had high accuracy based on test data [(mean AUC = 0.965), standard deviation 0.016]. A jackknife test revealed that the environmental variable with the highest gain when used in isolation was annual maximum water depth, which therefore appeared to have the most useful information by itself. The environmental variable that decreased the gain the most when omitted was vegetation, which therefore appears to have the most information that is not present in the other variables. The pre-survey model generated 28,665 hectares of potential suitable habitat across the study area, 5562 of which were in BICY (Figure 2). 

#### 2.1.2. Post-Survey Modeling

The post-survey model had a high accuracy based on test data [(mean AUC = 0.953) standard deviation 0.028]. A jackknife test revealed that the environmental variable with the highest gain when used in isolation was mean annual hydroperiod, as opposed to annual maximum water depth in the pre-survey model. The environmental variable that decreased the gain the most when omitted was again vegetation. The post-survey model generated 29,757.39 hectares of potential suitable habitat across the study area, 11,651.61 of which were in BICY. 

The final model output, which included the average plus the standard deviation to allow for a more liberal estimate of potential habitat generated 34,069 hectares of potential suitable habitat across the study area, 15,015 of which were in BICY (Figure 2). In the post-survey model, 64 of the 73 subsp. *austrofloridense* points were within the model output, a vast improvement over only six individuals in the pre-survey models. Based on the large spatial area classified as suitable habitat, along with the fact that 85% of individuals identified to subspecies revealed to be subsp. *austrofloridense*, we estimate the subsp. *austrofloridense* population in BICY to be between 1000–10,000 individuals.

### 2.2. Surveys

Over the course of the project, we surveyed over 215,000 m or 133.5 total miles based on track log data. Based on this, our spatial extent of detailed rare plant surveys was just over 80 hectares, representing less than 1% of the potential suitable habitat. We recorded 245 separate points of *S. reclinatum* s.l. in the areas we surveyed. From these, we subsampled the leaves of 96 individuals. 

### 2.3. Microscopy 

Evaluation of the leaf samples revealed that at least 73 (85% of specimens) of the individuals we sampled were subsp. *austrofloridense*, and 13 (15% of specimens) were subsp. *reclinatum* (see Figure 2 for collection locations, and Figure 3 for images).

While abaxial laminar pubescence varied with individuals and leaf maturity, we did not observe a single specimen displaying the characteristic rufous, tomentose pubescence of subsp. *austrofloridense* plants in rockland habitats of Miami-Dade County. It is worth noting that every specimen we sampled south of Loop Road was identified as subsp. *austrofloridense*. However, *S. reclinatum* s.l. for unknown reasons was far less frequently encountered in the Lostmans Pines region. 

Plants were particularly abundant along the rock reef formation in the middle of the Loop Unit, running east from Frog Hammock Camp. Generally, plants were found in “thickets”, or areas characterized by high coverage of other shrubs or small trees relative to the broader landscape. These conditions were found in a variety of cypress, pineland, prairie, and ecotonal habitats. By habitat, *S. reclinatum* s.l. was found most commonly in pineland/prairie ecotone (*n* = 96), followed by: cypress/prairie ecotone (*n* = 76), pineland (*n* = 43), pine/cypress/shrub mix (*n* = 15), marl prairie (*n* = 12), and cypress dome (*n* = 1). With the exception of marl prairie, where only subsp. *austrofloridense* was found, at least one of each subspecies was documented in each of the above listed habitat types, making definitive statements about habitat preference challenging. However, there was a trend toward subsp. *austrofloridense* in BICY being found in habitats with longer hydroperiods when compared to subsp. *reclinatum* (Table 1). It is important to note that 77.0% of subsp. *reclinatum* specimens were found in pine habitat or pineland/prairie ecotones, while this was only 39.7% for subsp. *austrofloridense*. Similarly, 57.5% of subsp. *austrofloridense* specimens were found associated with pond cypress (*Taxodium ascendens* Brongn.) habitat (pine/cypress or cypress/prairie), while this was only 23.1% for subsp. *reclinatum*. The specimen found in the cypress dome was not identified to a subspecies. 

## 3. Discussion

Our results represent a significant range extension for the federally threatened subsp. *austrofloridense*. A specimen from Monument Lake was formerly the northernmost known station (Sadle, 630; Bradley, 1547), yet we collected a sample that was identified as subsp. *austrofloridense* outside of the Big Cypress Institute, roughly 16.9 km to the northwest of Monument Lake. Perhaps the most compelling (and confounding) finding from this study was the confirmation of many new occurrences of subsp. *austrofloridense* in BICY, while also confirming population-level sympatry with subsp. *reclinatum*. We found that several populations were mixed, at times with plants just meters from one another (see Figure 2), and despite all plants superficially resembling subsp. *reclinatum* (i.e., lacking abaxial laminar pubescence), the vast majority of individuals sampled matched the micromorphological character of subsp. *austrofloridense*. 

We found a tendency toward subsp. *austrofloridense* being more commonly found in longer hydroperiod microsites associated with pond cypress relative to subsp. *reclinatum,* but recognize that the low sample size of the latter restricts this analysis. This result is counterintuitive, since a more sculptured laminar surface, such as found in subsp. *austrofloridense*, is more typically an adaptation to hotter, drier climates [18]. Evolving in rockland soils, such as present in southern BICY and Everglades National Park where subsp. *austrofloridense* occurs, that lack of the capillary capacity to remain saturated during the dry season would favor such adaptations, along with increased pubescence to prevent water loss. 

We recognize several limitations in our modeling approach that should be considered in interpretation thereof. For one, our environmental variables were highly correlated. Factors such as hydrology, fire-return intervals, and soil type greatly influence vegetation type, making interpretation of individual variable contributions challenging [19,20]. Secondly, the geographic scope of our models was limited by the geographic extent of long-term hydrologic data provided by the United States Geological Survey’s (USGS) Everglades Depth Estimation Network (EDEN). Additionally, the resolution of our models (50 × 50 m) fails to capture factors such as microtopography (e.g., solution holes or small outcrops) and higher resolution vegetation associations such as “thickets” and ecotonal habitats that are likely to influence the occurrence of subsp. *austrofloridense*. Despite these limitations, we believe that our post-survey HSM represents the best available estimate of potentially suitable habitat for subsp. *austrofloridense* and can be used as a foundation for further, more detailed spatial research as well as a baseline tool for conservation biologists. 

This study also highlights the importance of the use of micromorphology in taxonomy, specifically in the identification of cryptic taxa and divergent traits. In our study, this attention to detail provided improved clarity into the spatial distribution of a federally listed taxon. We found that while SEM provides far-superior imagery, at least in this case, the relative cost and speed of processing samples should be considered when choosing a methodology. In this case, we found that a trained observer using the high-powered dissecting microscope could identify a leaf sample with confidence in a matter of seconds, despite the relatively low quality of the captured image, and so can quickly abandon the expensive and time-intensive SEM method. Most field biologists likely can get access to a high-powered dissecting microscope through a local university or research institution for a reasonable expense and thus should not be intimidated by the prospects of this level of detail in their work, if deemed necessary. 

Our work has determined that these two otherwise geographically isolated taxa overlap in BICY, at the edge of their respective ranges in southwest Florida, and may in fact be actively hybridizing, with intraspecific introgression driving trait expression throughout the hybrid zone. Harrison and Larson [21] discuss the “semi-permeability” of the species boundary and outline analyses of the extent of introgression and interpretation of observed patterns in hybrid zones. Generally, these analyses take place on a geographical or ecological “cline”, e.g., latitude, precipitation, etc., through which selective pressures shape allele and genotypic frequencies. These clines can be narrow, as has been documented in *Artemisia tridentata* Nutt. subspecies in Utah where the “basin” and “mountain” taxa generate distinct hybrids across a range roughly 40 m in elevation that occurs rather abruptly in the landscape [22]. However, in the case of *S. reclinatum* s.l. the range of pineland and marl prairies (at times overlying exposed limestone) broadly spans multiple kilometers across the BICY landscape, and thus a broad “hybrid zone” where taxa express superficially similar macro traits should be expected. Natural hybridization can increase intraspecific genetic diversity, and lead to increased potential for adaptation to environmental change [23], and thus it is important to protect this natural hybrid zone. Furthermore, it would not be surprising to find subsp. *austrofloridense* even further to the north, perhaps even moving north with climate change over time if it is in fact more adapted to warmer temperatures and more pronounced dry seasons. 

## 4. Materials and Methods

### 4.1. Study Site

Big Cypress National Preserve (BICY) consists of 295,000 hectares made up primarily of cypress swamp, pinelands, and marsh communities located in Collier, Monroe, and Miami-Dade Counties in southwest Florida. Topography is relatively flat, gently sloping in a generally southwest direction toward sea-level [24]. The climate has been classified as tropical savanna, with hot, humid summers characterized by relatively high precipitation, and mild, dry winters [25]. The pronounced seasonal variation in precipitation leads to periods of shallow sheet flow across the landscape during the summer and fall. Sheetflow subsides following the rainy season and standing water is found only in deeper slough habitats. Despite a low-relief landscape, a patchwork of habitats is expressed largely based on subtle changes in elevation that determine the hydroperiod [24]. Generally speaking, the lowest areas contain cypress swamps, which transition to marsh habitats at moderate hydroperiods, with pinelands at the highest elevations. 

In the southeast portion of BICY where most of our surveys took place, the Pliocene, quartz-rich limestone bedrock is very close to the surface and sometimes exposed, particularly in pinelands, earning them the moniker of pine rocklands. These pine rocklands have a characteristic savannah-like canopy of slash pine (*Pinus elliottii* Engelm.) with understories dominated by saw palmetto (*Serenoa repens* (W.Bartram) Small) with a diverse suite of graminoids and tropical and temperate forbs [26]. These pinelands in BICY typically flood for a short portion of the year [27]. Most of the marshes in this area are marl prairies, diverse, low-stature graminoid communities with short hydroperiods and calcareous marl soils [26]. Similar habitats exist in the Everglades National Park, yet the pine rocklands there are slightly higher, and thus rarely flood and are on a ridge of younger limestone from the Pleistocene called the Miami Rock Ridge [27]. These pine rockland/marl prairie communities are unique to South Florida and boast a high degree of endemism [28]. 

### 4.2. MaxEnt Modeling

For the initial habitat suitability model, we used subsp. *austrofloridense* occurrence data from IRC and Corogin and Judd [14,15] within BICY, and occurrence data generated by IRC from EVER. When occurrence data were in the form of a polygon, we used a 25 × 25 m fishnet in ArcMap to generate points within the polygons. For the post-survey model, we included the new occurrences documented by Fairchild Tropical Botanic Garden (FTBG) in this study.

We created raster layers for environmental variables from polygon layers of vegetation, soils, and fire frequency (see Table 2). We also worked with Brian McCloskey from the United States Geological Survey’s Everglades Depth Estimation Network (EDEN) to generate raster layers of decadal means (2012–2021) of annual discontinuous hydroperiod (count of all the days in the climatic year that have water depth > 0 cm above ground surface), wet season depth (mean depth 1 June–31 October), and dry season depth (1 November–31 May). All raster layers were assimilated to cells of 50 × 50 m within the study area. The limiting factor of the study area extent was the EDEN network footprint, which is intended to cover freshwater habitats of the Everglades region. Note that the footprint of the EDEN network does not cover the entirety of BICY or EVER, but does cover all relevant areas of pine rockland and marl prairie in which subsp. *austrofloridense* has ever been known to occur, including the Lostmans Pines area, the Loop Unit and limited areas north of Tamiami Trail in BICY. The EDEN network covered all known subsp. *austrofloridense* populations in the Everglades National Park. Several small, isolated populations of subsp. *austrofloridense* occur in urban preserves of Miami-Dade County, but were not included here due to the inability to model hydrology. 

We generated models using a maximum entropy approach in MaxEnt (version 3.4.4; http://www.cs.princeton.edu/~schapire/maxent/ accessed on 8 December 2021 [19,20]). For each run we used 5000 maximum iterations with 10,000 maximum number of background points and a convergence threshold of 0.00001. We subsampled 25% of the occurrence data as test data, leaving the other 75% for training the model. For each final run we used 25 replicates. Since the output of MaxEnt is a continuous probability field, we determined the suitable habitat threshold from each model using maximum training sensitivity plus specificity as suggested by Jiménez-Valverde et al. [29] and utilized by Oleas et al. [30]. Maximum training sensitivity minimizes false negatives and specificity targets a reduction in false positives. We assessed the performance of each model by a receiver operating characteristic analysis (ROC), averaged over replicate runs, using the area under the curve (AUC) of test data (i.e., Test AUC), with X axis as 1-specificity, and sensitivity (1-omission rate) on the Y axis. We assessed the contribution of each environmental variable with a jackknife analysis generated by MaxEnt, wherein the test gain for each variable is given both without said variable and with only said variable, which can be compared to the test gain where all variables are used.

### 4.3. Surveys

We conducted field surveys between January and March of 2022 on eight separate days. Biologists from FTBG were assisted by experienced FTBG volunteers as well as BICY staff, totaling approximately 300 person hours of surveys across the eight days. We prioritized areas where models predicted habitat was suitable, but which had not yet been surveyed by previous researchers. Each surveyor was equipped with a smart phone with Avenza Maps software (Avenza Systems, Toronto, ON, Canada) and a series of georeferenced HSMs in PDF format, at various scales to be able navigate and record track and point data offline in remote parts of BICY. A schema was generated within Avenza with dropdown tabs for taxon, number of individuals, habitat, etc., to maintain consistency in data collection between surveyors. When surveying in teams, individuals would spread out by a minimum of 10 m. To estimate a range of total spatial coverage of surveys, we applied a conservative five-meter buffer to all track logs to account for detectability in a variety of habitats, deleting all duplicate tracks along more frequently used access roads and trails.

When a *S. reclinatum* s.l. individual was located, surveyors collected a point, along with species name, number of individuals, habitat, observer name, presence of standing water, sample number (if collection was made), along with any additional notes or photographs. Since reliable identification characters of subspecies were known to be microscopic, we collected multiple leaves from a subset of individuals (n = 96) across the range of the survey area for later determination. Leaves were placed in small coin envelopes and given a sample number. At the close of each day, envelopes were placed into a plant press and were later dried before analysis.

### 4.4. Microscopy

A combination of scanning electron microscopy (SEM) and high-powered dissecting scopes (1000×) was employed for examination of abaxial leaf surface micromorphology. All leaf material collected during our field surveys was pressed and dried before examination. Specimens were identified to subspecies following [12] wherein: subsp. *austrofloridense* “epidermal cell outlines always clearly visible and marked by an impressed groove. Surface elaborately ornamented with reticulate pattern in strong relief”, and subsp. *reclinatum* “epidermal cell outlines not always clearly visible. Surface generally smooth and irregularly undulating”. Microscopy imagery from that publication were referred to as a guide.

#### 4.4.1. SEM

Leaf material was mounted on carbon adhesive tabs on aluminum specimen mounts. Samples were rendered conductive by coating them with a gold-palladium alloy in argon vacuum for 90 s using a SPI-MODULE sputter coater (Structure Probe, Inc., West Chester, PA, USA). Samples were examined with a JEOL JSM 5900LV low vacuum SEM (SEMTech Solutions, Inc., North Ballerica, MA, USA), which includes X-Stream Imaging System to acquire digital micrographs. SEM work was conducted at the Florida Center for Analytical Microscopy at Florida International University. A total of thirteen (13) specimens were examined using this method, and images were processed at 500× magnification.

#### 4.4.2. Digital Microscopy

A total of 83 specimens were examined using this method, and images were processed at 1000× magnification using a Keyence VHX 1000 (Keyence Corporation of America, Itasca, IL, USA) digital microscope with integrated charged-coupled device (CCD) camera. At this high level of magnification, the operator could pan and use the auto-focus function to view multiple sections of leaf, but could often only get a portion of photos in clear focus. Since the goal was to examine as many leaves as possible and the focus was on identification and not final images, we did not stack images for a clearer final product, admittedly leaving portions of photos in poor focus. This work was conducted at the Florida International University Trace Evidence Analysis Facility.

## 5. Conclusions

This study has revealed the geographic range of subsp. *austrofloridense* to be far more extensive than previously known. As a result, federally designated critical habitat may need to expand. Sympatry of both subspecies, along with seemingly intermediate forms differing merely by cryptic morphological differences suggests potential for an active hybridization zone. Thus, drawing clear boundaries for each subspecies, i.e., putative parents versus hybrids, etc., will remain a challenge. Despite this, the hybridization zone warrants protection as hybridization is an important mechanism in plant evolution [21,23]. Future efforts in defining and conserving these taxa should include additional field surveys and genetic analyses to determine the degree to which the micromorphological differences correspond with genetic differences, and if in fact the genetic differences are significant enough to warrant taxonomic recognition for cryptic BICY populations of subsp. *austrofloridense*. Consideration of how to deal with potential intraspecific hybrids or introgressed populations should be considered from both a taxonomic and regulatory perspective. Genotypic studies within BICY could make for a fascinating story in active evolution taking place in South Florida and should be pursued to better understand gene flow between the two taxa and overall rarity of subsp. *austrofloridense* to inform management and conservation priorities. 

## Figures and Tables

**Figure 1 plants-12-01430-f001:**
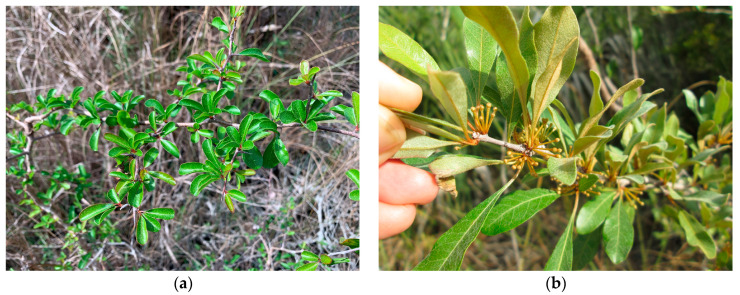
In situ specimens of *S. reclinatum* (**a**) superficially resembling subsp. *reclinatum* with glabrous surfaces from Big Cypress National Preserve, and (**b**) characteristic subsp. *austrofloridense* with rufous-tomentose abaxial surfaces, and pubescent pedicels and calyx from Everglades National Park.

**Figure 2 plants-12-01430-f002:**
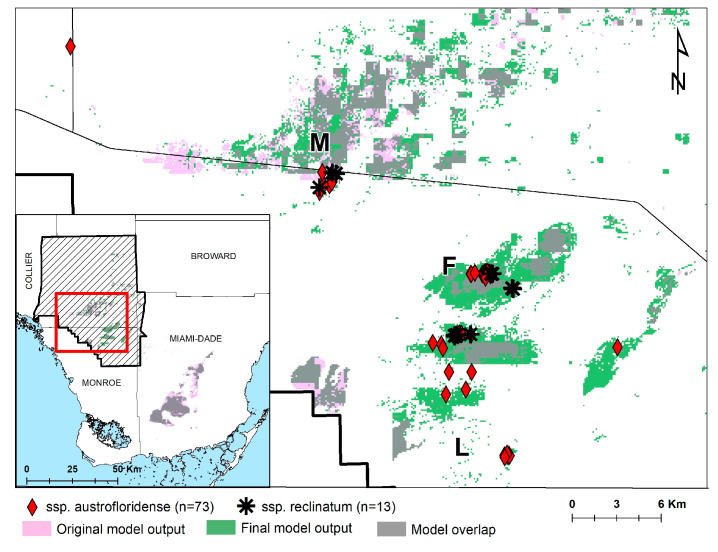
Approximate locations of *Sideroxylon reclinatum* s.l. found during the study that were identified to subspecies. Subsp. *austrofloridense* is represented by red diamonds and subsp. *reclinatum* is represented by black asterisks. Green polygons represent the final HSM model output for focal areas of BICY (map, extent represented by red square in inset) and for the entire study area (inset). BICY boundary is represented as black crosshatch in the inset. Landmarks discussed in results are displayed on the map: M = Monument Lake, F = Frog Hammock, L = Lostmans Pines.

**Figure 3 plants-12-01430-f003:**
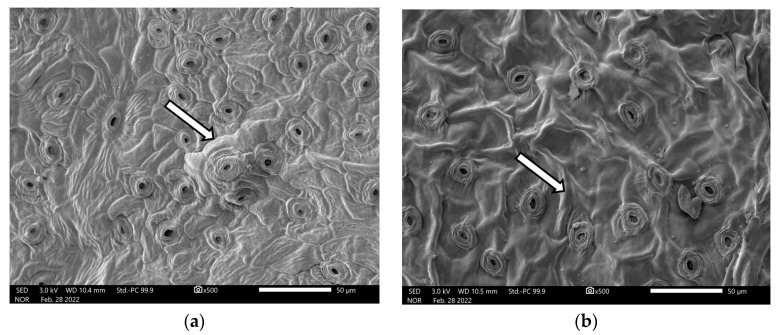
Comparative scanning electron microscope (SEM) imagery of abaxial laminar surfaces of separate specimens of (**a**) *S. reclinatum* subsp. *austrofloridense* and (**b**) *S. reclinatum* subsp. *reclinatum*. Note the specimens on the left display distinct epidermal cell outlines marked by impressed grooves (arrows), and that specimens on the right display less-distinct cell outlines and exhibit ridges that lack impressed grooves (arrows). See Appendix A for digital microscopy images.

**Table 1 plants-12-01430-t001:** Habitats in which *Sideroxylon reclinatum* s.l. specimens that were identified to subspecies were found using the percentage of each taxon’s conclusive occurrences. Habitats with a “/” represent ecotonal, or mixed associations. The table represents a rough gradient of increasing hydroperiod from left to right.

Subspecies	Habitat				
	Pine	Pine/Prairie	Prairie	Pine/Cypress	Cypress/Prairie
*austrofloridense*	26	13.7	2.7	16.4	41.1
*reclinatum*	46.2	30.8	0	7.7	15.4

**Table 2 plants-12-01430-t002:** Data layers used for MaxEnt modeling with geographic extent, source, year(s), file type, and additional comments. * The spatial extent of these layers was limited to that of the USGS EDEN Network.

Data Layer	Geographic Extent	Source(s)	Year(s)	File Type	Description
*Sideroxylon* occurrences	BICY and EVER	IRC	2012–2014	Point	Includes field data points as well as XY points on 25-m grid that intersect ENP occurrence polygons. One point near Monument Lake Campground was obtained from verified specimen record in Corogin and Judd 2018.
		Corogin and Judd	2014		
		FTBG	2022		
Mean Annual Hydroperiod	BICY and EVER *	EDEN	2012–2021	Raster	Annual Discontinuous Hydroperiod (1 May–30 April climatic year; count of all the days in the climatic year that have water depth > 0 cm above ground surface; averaged across one decade (2012–2021).
Mean Wet Season Depths	BICY and EVER *	EDEN	2012–2021	Raster	Average Wet Season Water Depth (1 June–31 October) over one decade (2012–2021)
Mean Dry Season Depth	BICY and EVER *	EDEN	2012–2021	Raster	Average Dry Season Water Depth (1 November–31 May) over one decade (2012–2021).
Annual Maximum water depth	BICY and EVER *	EDEN	2012–2021	Polygon	Average Annual Maximum Water Depth (1 November–31 May) over one decade (2012–2021).
Vegetation	BICY and EVER	NPS	2017, 2020	Polygon	Level 6 Classification was selected for this model.
Soils	BICY and EVER	USGS	1948, 2012	Polygon	Soils were dissolved based on soil unit name.
Fire Frequency	BICY and EVER	NPS	1978–2020	Polygon	This raster was created by overlapping all fire polygons in the two parks and obtaining the number of fires for each specific area and dividing it by the number of years (42, 1978–2020).

## Data Availability

All data generated for this project can be made available upon request to Big Cypress National Preserve. Please contact Courtney Angelo at courtney_angelo@nps.gov.

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
