# Peer review of "Population Assessments of Federally Threatened Everglades Bully in Big Cypress National Preserve, Florida, USA, Using Habitat Suitability Modeling and Micromorphology"

_plants, 2023, doi:10.3390/plants12071430_

Round 1
Reviewer 1 Report
This manuscript reports a study on a federally threatened plant in the Big Cypress National Preserve in the USA. The authors conducted thoughtful field searches guided by maps generated by habitat modeling. In addition, they carried out laboratory work on leaf samples using three different imaging tools, i.e. high-powered dissecting scopes, SEM and Digital microscopy to generate leaf epidermal cell images to compare these three different approaches in distinguishing the target subspecies (Sideroxylon reclinatum ssp. austrogloridense) from its type species (Sideroxylon reclinatum ssp. reclinatum). These laboratory work not only inform the best tool in distinguishing the subspecies it also provided more data to refine the species range modeling. The outcome of this well thought out research provides very valuable information to the managers of this rare and threatened species to make better conservation plans. As a rare species conservation researcher, I consider this study an example of how rare species conservation should be done. I have only a few minor suggestions to change.
(1) The authors found/confirmed that the rare subspecies is sympatric with its wide spread subspecies, over a wide range in the Big Cypress National Preserve, and are in fact hybridizing. I think it is desirable to point out that this is a natural hybridization zone. Natural hybridization are likely increase intra-species variations and provide the species more adaptation potential for environmental changes. Therefore such natural hybridization zone should be protected. For references on contribution of hybridization to speciation, see Rieseberg (1997) Hybrid origins of plant species, in Annual Review of Ecology and systematics.
There are concerns about hybridization between rare and common species (including subspecies level), indicating potentially harmful consequences (Rieseberg and Geber 1995, Hybridization in the Catalina Island Mountain Mahogany (Cercocarpus traskiae), in Conservation Biology ). However, I believe that the Everglades bully situation in which the hybridization zone is natural phenomenon, and it should be leave along.
(2) The title and/abstract should indicate the study is in the USA – to inform international readers.
(3) Line 32, delete “limits of”
(4) Line 49, change “project” to “study”
Reviewer 2 Report
Ecotypic variation is common in plants, and only sometimes gets taxonomic recognition. Whether or not this distinction is warranted is another issue, but this reviewer sides with the taxonomic lumping especially as there appear to be some intermediates. But that aside:
The authors have written a clear and interesting manuscript, and their modeling approach a powerful tool for predicting where the endangered taxon might occur. I have only a few little comments on the manuscript, that I have made using the comments tool in Adobe.
In particular, I do not like the use of “SRA” for the taxon, and suggest they use a more botanically acceptable abbreviation. I think that “subspecies austrofloridense” or “ssp. Austrofloridense” will be more easily read and understood and will not take up that much more space!
Also I wonder if the researchers can posit why the micromorphological differences correspond with wetter habitats? Do the extra grooves make cells more flexible? It might be worth comparing this species with other taxa known to grow in drier and wetter habitats (e.g. Piriqueta caroliniana) to see if others have examined morphological differences that correspond to subspecies.

Reviewer 3 Report
The manuscript assessed the populations of a federally threatened plant (Sideroxylon reclinatum subsp. austrofloridense) in one of the USA national preserves. This subspecies is sympatric with its relative but much more widespread taxon (Sideroxylon reclinatum subsp. reclinatum). However, both are very hardly distinguishable and the authors used a combination of methods in order to identify the specimens and to evaluate the distribution and the population size of the threatened subspecies. They developed habitat suitability models. Habitat modeling is predicting species geographical distribution based on environmental conditions of known sites. The subject of this research is very important considering also that the spatial representation of habitat suitability is a good tool for supplementing conservation strategies.
Generally, I think that the topic of the paper is very interesting. In my opinion, the overall impression of the manuscript is good. It is well written, the research is well organized and executed. The introduction is informative, presents the problem undertaken and define the purpose of the work. The methodology and approach to the topic are good, and results are clearly presented. However, and after reading the paper, I have found some flaws which require to be resolved. They are as follows:
1. Lines 24-26: I suggest to write only keywords that are not present in the Title.
2. When you write the abbreviation of "sensu lato" (namely "s.l."), you use both "s.l." and "s. l." (with blank space) in the text and in the figure captions. You should be consistent throughout. I would suggest "s.l." as more commonly accepted in the botanical scientific articles, but it depends on you.
3. When you use the taxonomical category "subspecies", use the abbreviation "subsp." or "ssp." rather than the whole word "subspecies". For example, instead of S. reclinatum subspecies reclinatum (line 64) write S. reclinatum subsp. reclinatum. See also line 109, lines 112-114 (the Fig. 1 caption), line 133, line 134, line 137, line 164 (Fig. 2 caption), line 175, line 177, line 180, line 318, line 320. Whatever you choose ("ssp." or "subsp."), be consistent throughout. In my opinion, however, "subsp." is better option.
4. I am glad that you added the authority/authorities of the species or taxa included in the manuscript. I assume that you have used IPNI for the authorities, or at least POWO. However, some names that have initials are written in the MS with blank spaces (between the initials and the family names). They should be corrected in the text. I noticed two cases, e.g., Sideroxylon thornei (Cronquist) T.D. Penn. (correct to "T.D.Pann.", line 71); Serenoa repens (W. Bartram) Small (correct to "W.Bartram", line 243).
5. Line 90: "… area under the curve (AUC)scores …" to become "… (AUC) scores…".
6. Table 1, line 145: The abbreviation SRR is appearing here for the first time. I understand what it means, however you should explain in the figure caption both abbreviations. Each caption should be understandable without referring to the main text.
7. Compare the highlighted texts (in the attached file) in lines 94 and 147. Equalize.
8. Line 184: do you mean " fire-return"?
9. Line 213: add the authority of Artemisia tridentata.
10. Line 242: correct to "elliottii" (with double "t")!
11. The numbers of all subsections in Materials and Methods should be corrected. Now you have "4.1." twice (see lines 226 and 253). Renumber all subsections thereafter.
12. Line 255: Corogin & Judd are Reference [15]. Add the corresponding number in the square brackets.
13. Line 258: specify the abbreviation FTBG appearing here for the first time.
14. Now you have two "Table 1". Correct in line 260 and in the caption of the second Table – both are Table 2.
15. In the actual Table 2, the font size of "Sideroxylon occurrences" (the 1st row) seems different and should be increased.
16. Line 285: check the references [28] and [29]. Actually, there are only 28 references included in the list; and Jiménez-Valverde et al. and Oleas et al. are 27 and 28, respectively. Maybe you should check all the cited references in the main text.
17. In Materials and Methods: you didn't include the jackknife method in this section, however you use its results in the Result section. I assume that the test was used implemented in MaxEnt to indicate the relative contribution of each environmental variable in the model?
18. When you write the levels of magnification in the text [namely "500x magnification" (line 332) and "1000x magnification" (line 336)], use the symbol ×, not the letter "x"!
19. I definitely think that it would be good to present one or two photographs of the threatened Sideroxylon reclinatum subsp. austrofloridense, maybe the habit and/or some details of the gross morphology. Not only images of the epidermal features.
20. Figure 1 (page 5): Fig. 1c and Fig. 1d present the digital microscope images. I admit that I have no experience in this high-powered dissecting microscopy. I use only SEM and LM. I know very well that the operator can focus only on parts or layers of the samples when identifying the specimens. When he/she takes pictures, only portions of the leaf will be in clear focus. However, I would prefer these images 1c and 1d to be cropped and to present only the clear parts so the reader could see what is presented on and what you show with the arrows. The Figure 1 in the final production will not be very big in size. Now, I hardly see the grooves for example; I have to zoom very much in order to see some details. Furthermore, if parts of the images are cropped, you should draw the corresponding scales, which obviously won't be 100 µm.
21. Figure 2: explain in the caption what the red box and the shaded box on the left of the Figure 2 mean.
22. There are some references in the References list that need corrections. See the attached file.
Additional typos or comments are included in the attached pdf file. Please, see them.
